# Design and Analysis of the High-Speed Permanent Magnet Motors: A Review on the State of the Art

Qiping Shen *, Ziyao Zhou, Shan Li, Xinglin Liao, Tao Wang, Xiaorong He and Jingshan Zhang

School of Electrical and Electronic Engineering, Chongqing University of Technology, Chongqing 400054, China; 52200712112@stu.cqut.edu.cn (Z.Z.); lishan@cqut.edu.cn (S.L.); lxl023040@cqut.edu.cn (X.L.); wangtao20190031@cqut.edu.cn (T.W.); hexiaorong@cqut.edu.cn (X.H.); 52200712123@stu.cqut.edu.cn (J.Z.)
* Correspondence: shenqp1983@cqut.edu.cn

**Abstract:** This paper provides an overview of the design and analysis of high-speed PM motors by focusing on prominent issues such as motor losses, temperature rise, rotor strength and vibration. The design challenges of high-speed PM motors are briefly described, and the application of various stator and rotor structures and materials is presented in electromagnetic design. Combined with the temperature distribution of the motor, various heat dissipation measures to suppress the temperature rise are summarized. Strength and dynamics analysis of the rotor are outlined with respect to the safety of rotor operation. The current status of coupled multi-physics domain design used to improve the comprehensive design capability of high-speed PM motors is reviewed. Future directions of technologies related to the design of high-speed PM motors are pointed out.

**Keywords:** high-speed PM motor; electromagnetic design; temperature rise; strength and dynamics analysis; multi-physics domain





## 1. Introduction

High-speed motors have been developed over a long period of time and have been widely used in machine tools, turbochargers, mechanical turbo-charging systems, flywheel energy storage systems, gas compressors, blowers, vacuum pumps, shipborne power supply systems and aero-engines, etc. They typically operate at speeds in excess of 10,000 r/min, rotor circumferential speeds ($v_c$) in excess of 50 m/s and r/min$\sqrt{kW}$ (the product of speed and power square root) in excess of $1 \times 10^5$ [1–4]. In terms of structural principle, both DC motors and AC motors can realize high-speed operation. The main structural types include IM, PMSM and SRM. Referring to [3–60], the power and speed distributions of the current high-speed motor studies are given in Figure 1. IMs operate by the interaction of the induced current generated by the rotor winding in the stator magnetic field with the stator magnetic field. SRMs rely on the principle of minimum reluctance to generate torque, that is, the magnetic flux always closes along the path of minimum reluctance, thus creating a magnetic pull to rotate the rotor. PM motors are driven by the interaction of the magnetic field generated by the permanent magnets and the rotating magnetic field generated by the stator winding. The characteristics of these high-speed motors are shown in Table 1 [61]. As can be seen from Table 1, the high-speed PM motor has outstanding advantages at the efficiency and power density levels. Among them, the power density distribution of high-speed PM motors at different speeds and powers is shown in Figure 2. Moreover, compared with other motor structures, PMSM has the advantages of diverse stator and rotor structures, as well as characteristics of good control. Therefore, PM motors are more widely used in high-speed fields.

Figure 3a shows the status of the speed-power study for a variety of structures of high-speed PM motors. Detailed raw data are presented in Appendix A. The distribution of $v_c$ and r/min$\sqrt{kW}$ of high-speed PM motors is given in Figure 3b. As we can see from the figure, the circumferential speed $v_c$ of the SPM structure design is generally

higher than that of the IPM structures and cylindrical PM structures. Moreover, the SPM structure is easier to achieve higher power of 100 kW and above, as well as higher speed of 100 krpm and above. However, there are more challenges in electromagnetic design, thermal management, mechanical structure and dynamics analysis for high-speed PM motors compared with traditional low and medium-speed PM motors.

**Table 1.** Characteristics of various types of high-speed motors.

|  | High-Speed IM | High-Speed SRM | High-Speed PM Motor |
|---|---|---|---|
| Advantages | easily starting<br>low cost<br>rotor can withstand high temperature | simple rotor structure<br>low rotor loss<br>short end of winding | high efficiency<br>high power density<br>high power factor |
| Disadvantages | high rotor loss<br>low power factor<br>laminated rotor end rings are easily damaged | low efficiency<br>high noise<br>large wind friction | low rotor strength<br>permanent magnets are easy to demagnetize<br>higher cost |

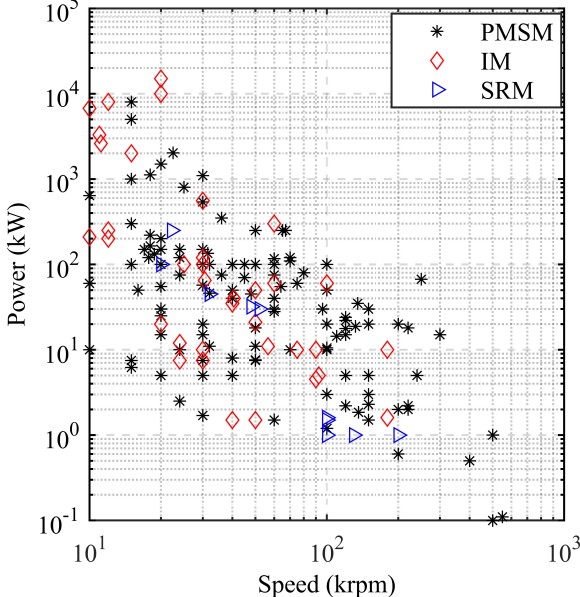

**Figure 1.** Speed and power distribution of high-speed motors of different structure types.

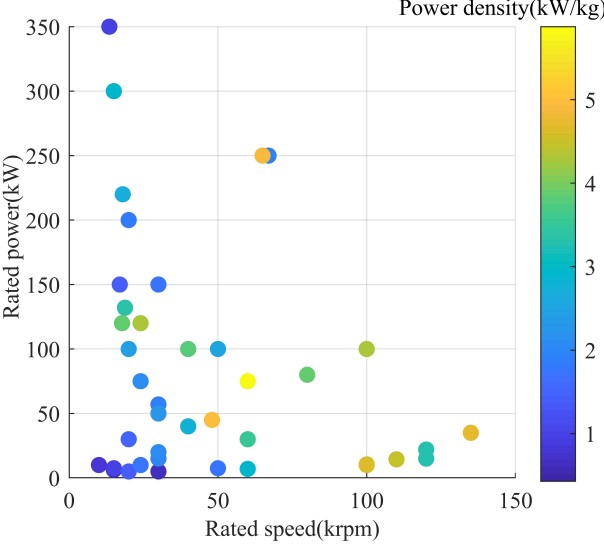

**Figure 2.** Distribution of power density of high-speed PM motors at different speed and power.

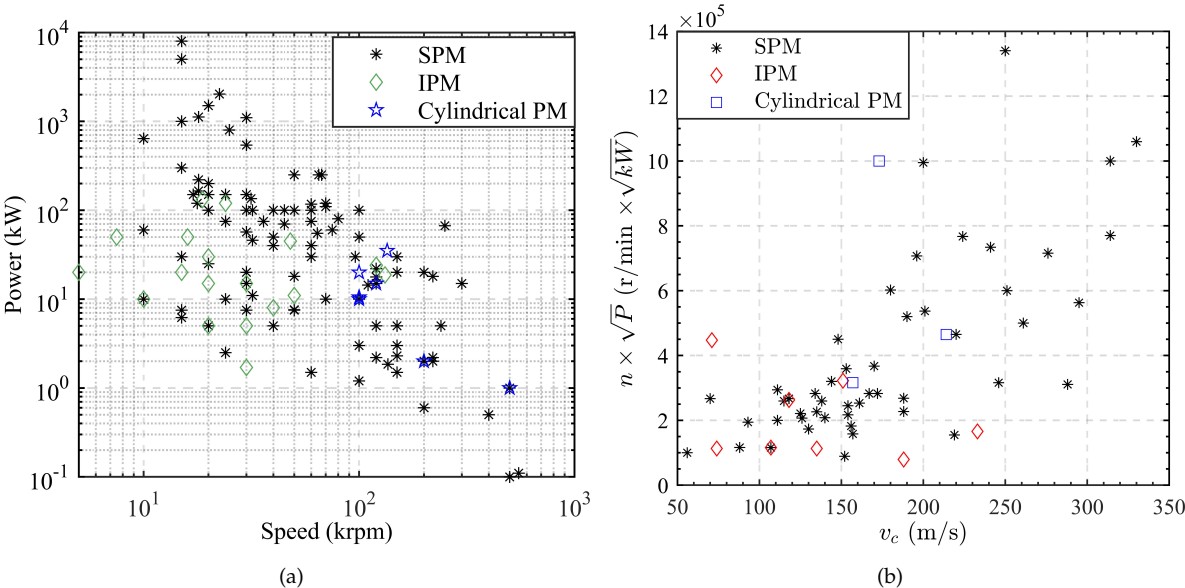

**Figure 3.** Power and speed distribution of high-speed motor: (**a**) Speed and power distribution of high-speed PM motors with different structures; (**b**) Distribution of $v_c$ and $\mathrm{r/min}\sqrt{kW}$ of high-speed PM motors.

Firstly, the frequency of high-speed PM motors is generally up to thousands of hertz, so the loss density and thermal effect of the motor is much higher than that of low and medium speed motors. To overcome these problems, it is necessary to innovate motor structures and to be more careful in the choice of materials compared to conventional motors [54].

Secondly, the high-speed PM motor is compact, while its wind friction loss and rotor eddy current loss are much higher than those of conventional motors, so the highest temperature region in the motor is shifted from the stator to the rotor [60]. Seeking thermal management measures to solve the efficient heat dissipation of the rotor is extremely important for the efficient operation of the motor.

Thirdly, the rotor circumferential speed ($v_c$) of high-speed PM motors can reach up to 200 m/s and above. The problem of rotor strength is prominent due to high-speed centrifugal force and thermal stress [62]. Finally, compared with conventional motors, the rotor of high-speed PM motors is more prone to bending vibration. To ensure the safe and stable operation of the motor, rotor strength analysis and calculation of critical speed are indispensable parts of the design of high-speed PM motors.

In summary, there are more factors to be considered at each design level for high-speed PM motors than for conventional low and medium speed motors. This paper first outlines the selection of stator and rotor structures and materials for high-speed PM motors from the perspective of improving electromagnetic performance, and then reviews the current status and key issues of research in four areas, namely thermal management, rotor strength, dynamics analysis and multi-physics domain coupling in turn. Finally, issues related to the further efficient design of high-speed PM motors are discussed.

## 2. Electromagnetic Design

The electromagnetic performance of high-speed PM motors is closely related to the structure and material of the motor. In order to avoid the harm caused by high loss, the structure and material of the stator and rotor need to be designed reasonably.

### 2.1. Stator Design

Stator design includes the selection of major dimensions and calculation of stator losses, as well as the structure and material selection of cores and windings. Stator design is one of the important links to reduce motor losses and improve motor stiffness.

The major dimensions of the motor can usually be estimated by (1).

$$D_{i1}^2 L_{ef} = \frac{6.1 \times 10^4}{K_\phi K_{dp} A B_\delta \alpha'_p} \frac{P'}{n_N} \tag{1}$$

where $D_{i1}^2 L_{ef}$ is the armature volume, $D_{i1}$ is the stator inner diameter, $L_{ef}$ is the armature calculation length, $\alpha'_p$ is the calculating pole arc factor, $K_{dp}$ is the winding factor, $K_\phi$ is the waveform coefficient of the air gap magnetic field, $A$ is the electrical load, $B_\delta$ is the air gap magnetic density, $AB_\delta$ is the Electromagnetic load, $P'$ is the calculated power, and $n_N$ is the rated speed.

Determining the major dimension ratio $\lambda$ is also a common method for estimating the overall size of a motor, which is defined as the ratio of the $L_{ef}$ to the pole pitch $\tau$, as shown in (2).

$$\lambda = \frac{L_{ef}}{\tau} \tag{2}$$

The selection of $D_{i1}^2 L_a$ and $AB_\delta$ for high-speed PM motors at different speeds and powers are given in Figure 4a,b, respectively. It can be seen that in the listed speed and power range, the $D_{i1}^2 L_a$ selection is mainly in the range of 2000 cm$^3$ and below, and there is a tendency to decrease with the increase in speed. The selection of $AB_\delta$ is mainly concentrated around $(0.5 \sim 3) \times 10^4$ A $\cdot$ T/m. Figure 5a,b show the selection of $\lambda$ and $D_{i1}/D_1$ for different speed and power, respectively. The larger $\lambda$ is, the slimmer the motor is, which helps to reduce the centrifugal force caused by high speed. However, a motor core that is too long will affect its operational stability. Choosing a relatively small value of $\lambda$ facilitates the placement of more permanent magnets to improve motor performance. $D_1$ is the outer diameter of stator. From Figure 5, we can see that the $\lambda$ is often taken in the range of $0.5 \sim 2.5$, and the $D_{i1}/D_1$ is mostly taken in the range of $0.3 \sim 0.8$.

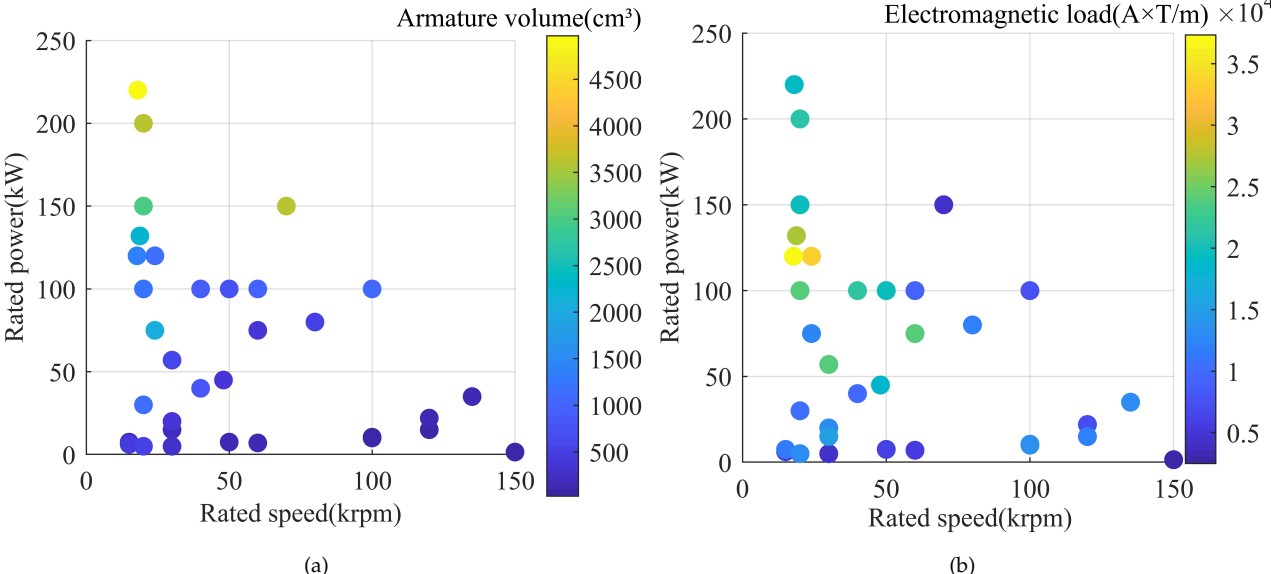

(a)  (b)

**Figure 4.** $D_{i1}^2 L_a$ and $AB_\delta$ distribution: (**a**) Distribution of $D_{i1}^2 L_a$ of high-speed PM motors; (**b**) Distribution of $AB_\delta$ of high-speed PM motors.

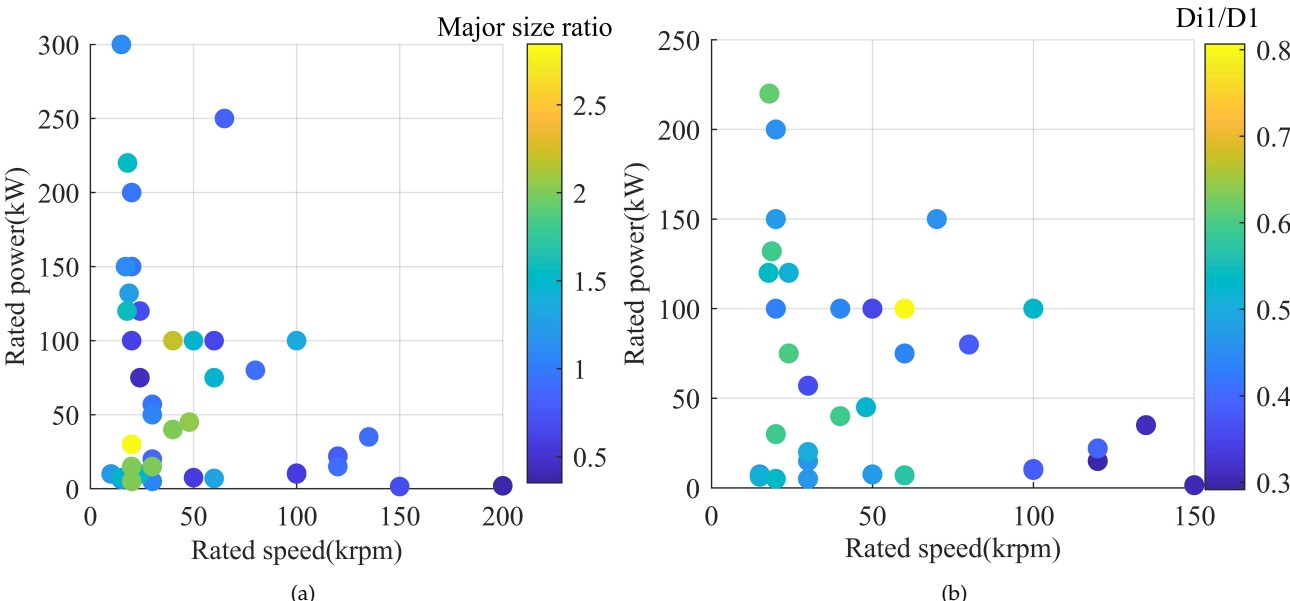

**Figure 5.** $\lambda$ and $D_{i1}/D_1$ distribution: (**a**) Distribution of $\lambda$ of high-speed PM motors; (**b**) Distribution of $D_{i1}/D_1$ of high-speed PM motors.

For the calculation of stator core losses. Generally speaking, the alternating frequencies of winding currents and core magnetic fields of high-speed PM motors are as high as thousands of hertz, which is likely to cause additional losses and serious temperature rise in motors. Figure 6 shows the loss curve of 35W270 silicon steel at different frequencies. It can be seen from Figure 6 that the iron loss at high frequencies is higher than that at low frequencies. Therefore, the accuracy of the early Bertotti iron loss separation model is greatly reduced. To accommodate high frequency operating conditions, Shanlin et al. [63] of the Harbin Institute of Technology propose a variable loss coefficient orthogonal decomposition core loss model, which can simultaneously consider the rotational magnetization in the core and the high-frequency skin effect, and is practical for high-speed motors with high fundamental frequency. The loss factor of the core has been shown to be related to the temperature rise and stress variation of the material as well [64]. In response, an improved model of iron loss in high-speed PM motors that can take various physical factors such as magnetic field, temperature and stress into account has been studied and applied [14,65].

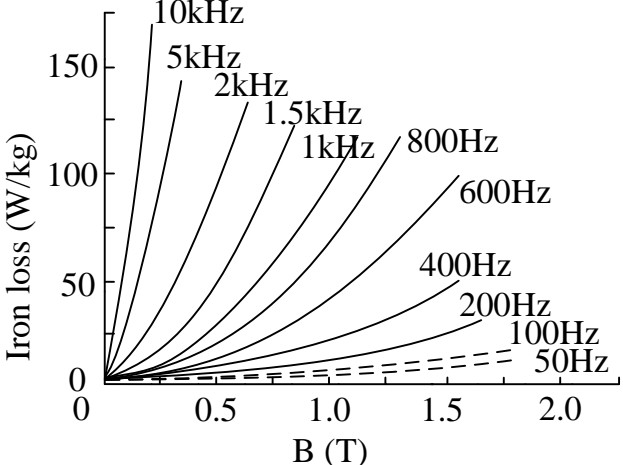

**Figure 6.** The loss curve of 35W270 at different frequencies.

High-speed PM motors are usually selected in 2 or 4-pole configurations, the purpose of which is to reduce the operating frequency, improve the sinusoidality of the current waveform and, in particular, reduce the eddy current losses in the permanent magnets. The common stator structures are few-slot, multi-slot, no-slot and fictitious slot, as shown in Figure 7.

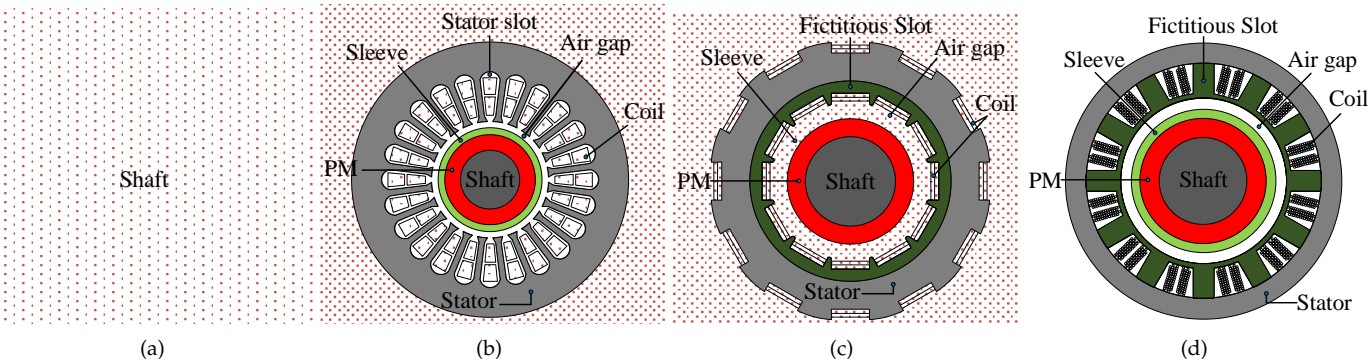

**Figure 7.** Stator configurations with different number of slots: (**a**) Few-slot structure; (**b**) Multi-slot structure; (**c**,**d**) is fictitious slots.

The slotted structure on the inner surface of the stator can achieve higher air-gap flux density and power density. However, stator slotting will cause changes in air-gap permeability and bring tooth harmonics, especially for the less-slotted structure shown in Figure 7a. However, due to the high torque density of this structure, it has some applications in low-power and lightweight high-speed PM motors [66]. The stator multi-slot structure shown in Figure 7b can reduce the cogging torque and weaken the harmonic air gap magnetic density. Meanwhile, measures such as adjusting the slot opening width of the stator [20], optimizing the pole-slot fit [53], and using the stator slant slot and magnetic slot wedge [67] can contribute to the weakening of motor eddy current loss and the improvement of material utilization.

On the contrary, the slotless structure has a stable air gap permeability, but the air gap magnetic density and torque of this structure is weak [68]. To make full use of the advantages of slotless and slotted structures, the virtual stator slot structures, as shown in Figure 7c,d, have gradually emerged in recent years to avoid the slot effect [69].

For the diversity of each stator structure, silicon steel with high silicon content and thickness between 0.2 mm and 0.5 mm is usually chosen as the core material, but the eddy current losses of silicon steel is obvious under high frequency conditions. With the improvement of process level, amorphous alloy material with lower hysteresis loss and thin strip thickness of 0.02~0.03 mm is gradually becoming the alternative material for traditional silicon steel in the high-speed field. However, the saturation magnetic density of this material is lower, while the more stress-sensitive nature of this material is not conducive to the low-vibration and low-noise operation of the motor. [70,71]. Another alternative material is the SMC, which has an insulating coating for each pure iron powder nanoparticle as illustrated in Figure 8a. Due to the special physical structure, the permeability of SMC is significantly lower than the first two materials, as shown in Figure 8b. Meanwhile, with the presence of the inter-particle insulation layer, the eddy current loss of this material is much lower than that of the silicon steel material in high frequency situations at upper kilohertz, and the material is easier to process to accommodate more complex core shapes [72–74]. Nanocrystalline materials with good magnetic properties at high frequencies are currently in the research and development stage and are expected to be widely used in the future [75].

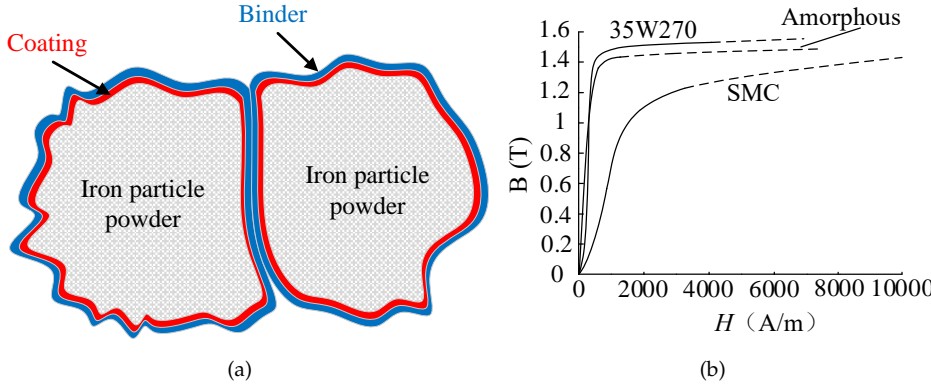

(a)　(b)

**Figure 8.** Material properties: (**a**) The basic components of soft magnetic composite [76]; (**b**) Magnetization curves of amorphous alloys, soft magnetic composite and silicon steel sheets [77,78].

At the winding configuration level, the concentrated winding structure is simple and compact with short end windings, which is usually used in some slotless motors or fields requiring compact structure. However, the Back-EMF waveform of this winding structure is not sinusoidal, and the rotor loss is large caused by high harmonics [79]. Compared to centralized windings, double-layer short-distance distribution winding can improve the Back-EMF waveform and magnetomotive force waveforms, thus improving electromagnetic performance, but the end winding of this structure is long [80]. The distributed double-layer toroidal winding is evenly distributed around the stator to produce a sinusoidal Back-EMF waveform. Meanwhile, the toroidal winding minimizes the length of the end winding as shown in Figure 9, which is conducive to the reduction in copper consumption and the improvement of motor stiffness [69]. Therefore, the toroidal winding structure is currently superior to other winding structures in high-speed, high-performance applications.

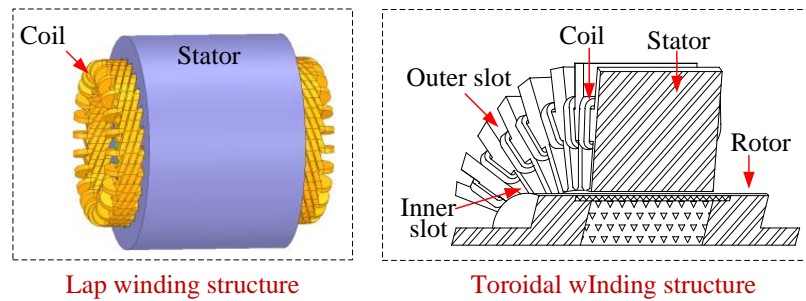

**Figure 9.** End length comparison between toroidal winding and conventional winding.

The winding losses in high-speed PM motors include basic copper consumption and additional losses due to skin and proximity effects at high frequency currents [68]. Figure 10 shows the winding losses at different frequencies and different wire diameters. Usually, the high-speed PM motors mostly use multi-stranded Litz wire as the stator winding to minimize the impact of skin effect [81]. Yet, the use of round Litzs wire generally requires a special process of dipping or lacquering, which increases the complexity of the process [82]. Another type of rectangular cross-section wire has a simple process, along with a better fill factor and lower Joule losses, but its eddy current loss is higher [83].

Copper is the most commonly used material for most windings. In pursuit of low loss and light weight of the windings, Volpe et al. [84] propose the use of aluminum instead of copper, but it was found in the experiments that the AC winding loss of the aluminum winding could not be reduced effectively. Currently, a team from Lappeenranta University of Technology has designed a carbon nanotube braided yarn based on nanotechnology. This material is extremely lightweight, has twice the electrical conductivity of copper-wound

wire, and is thermally stable. It also has half the heat loss of copper winding , which is expected to replace copper as winding wire in the future [76]. In addition, the composite technology of copper-CNT developed by Oak Ridge Laboratory provides more options for winding materials.

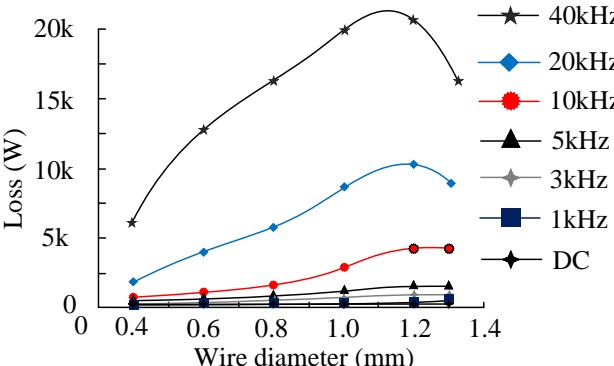

**Figure 10.** Winding losses at different frequencies and different wire diameters.

### 2.2. Rotor Design

Eddy current loss and wind friction loss in high-speed PM motors are prominent. The rotor eddy current loss is mainly caused by the time and space harmonics of the armature magneto-dynamic potential and the stator slotting effect [85]. For eddy current loss, there are usually two ways to calculate. The analytical method is suitable for eddy current loss calculations of simple structured rotors. The FEM, on the other hand, can take into account many nonlinearities at the same time and obtain more accurate calculation results, but the calculation speed is slower [86].

The protection of permanent magnets must also be taken into account at high speed. PM has the characteristic of pressure resistance but not tensile resistance, so the rotor of the high-speed PM motor often adopts the SPM structure with sleeve or the IPM structure.

For the SPM structure, fiber or alloy materials are mostly used to protect the magnets. The alloy material has better heat dissipation and stronger stiffness, but the harmonics of armature magneto-dynamic potential will generate large eddy current loss in the sleeve. On the contrary, although the fiber composite material is less effective in heat dissipation, it does not produce eddy current loss itself, so it is more likely to fulfill the low loss requirement of the rotor [87]. Both of these materials have non-permeable properties, which can be detrimental to the increase in magnetic load. Yon et al. [88], at the University of Bristol, designed a novel PM rotor sealing technique using semi-conducting materials as a protective sleeve to increase the motor magnetic load and counter potential fundamental wave amplitude by 20%.

In fact, the eddy current losses in SPM rotors are mainly concentrated on the surface of the PMs. The reduction in rotor surface loss can be achieved by adding shielding [89], adjusting the shield size [80,90] or by taking measures such as magnet segmentation, sleeve segmentation [67,87] and sleeve slotting [43]. When the permanent magnets are segmented, the air gap magnetic field is difficult to meet the design requirements of the sine wave. In response, reference [91] proposes a structure in which samarium cobalt and ferrite are placed in blocks and mixed along the circumference, as shown in Figure 11a, and indicates that this structure can improve the sinusoidal distribution of the air gap magnetic density waveform.

However, for high-power and high-speed PM motors, the segmentation of the magnet and sleeve will reduce the rotor stiffness to a certain extent, which is not conducive to high-speed operation. Additionally, the shield will no longer have a significant effect on eddy current suppression in high power and high speed motors. Reference [41] combined the advantages of low conductivity of ferrite and high remanence of rare-earth PMs, and proposed a composite excitation structure, as shown in Figure 11b. SmCo is used to

improve the magneto-dynamic potential, while ferrite is mainly used to suppress rotor eddy currents instead of shielding. The eddy currents on the surface of this composite excitation structure are significantly reduced compared with that of a single-excited rotor, as shown in Figure 12.

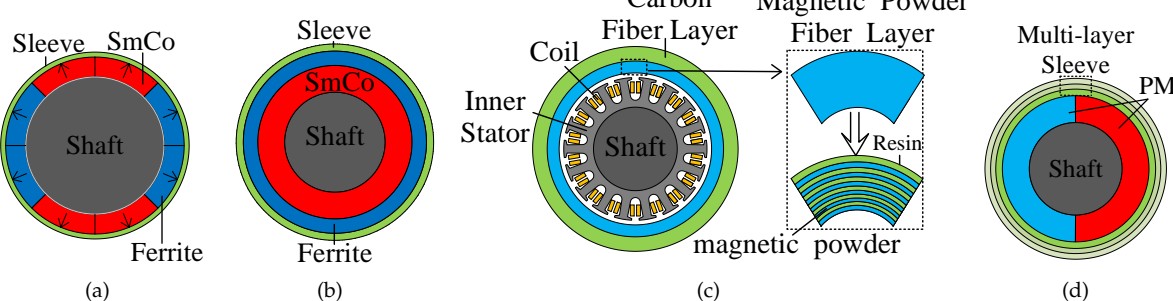

**Figure 11.** Rotor structures of high-speed PM motor with sleeves: (**a**) Circumferential composite excitation structure; (**b**) Radial composite excitation structure; (**c**) Magnetic powder fiber composite rotor structure; (**d**) Multi-layer sleeve mating structure.

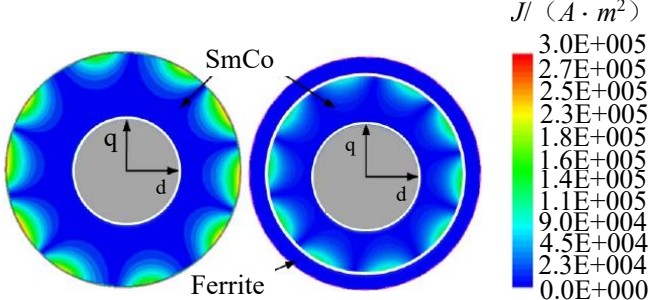

**Figure 12.** Comparison of eddy current distribution between single excitation and combined excitation [41].

To further reduce the rotor eddy current loss, Jingyue et al. [92] mixed NdFeB magnetic powder with epoxy resin and cured magnetization with reinforcing fiber based on the outer rotor structure, as shown in Figure 11c. The sleeves' layered coordination structure as shown in Figure 11d has also been further researched and applied in recent years. Its function is to reduce the thickness of the original sleeve by the interference fit of the multi-layer thin sleeve, which increases the physical air gap and greatly reduces the eddy current loss of the sleeve without losing rotor strength [93].

Compared with the SPM strucure, the poles of the IPM strucure are less influenced by the armature magnetic field, and its overload capacity and the capacity of PM resistance to demagnetization are better. However, the problem of stress concentration in isolated magnetic bridges has to some extent made them less used in high-speed applications [7,12]. The PM of the IPM structure is mostly segmented and circumferentially layered to improve rotor strength and suppress eddy currents inside the permanent magnet, as shown in Figure 13a. The interior tangential rotor structure shown in Figure 13b is also usable in high-speed applications, which is able to reduce the leakage of the IPM structure while providing a larger reluctance torque and reducing the number of magnets [94]. Figure 13c shows a non-uniform air-gap IPM (NUA-IPM) structure. The reverse "V-shaped" magnet improves the pole arc coefficient of the magnet, and by adjusting the non-uniform air gap, the harmonic amplitude in the air gap magnetic density waveform can be reduced, but the air friction loss of this structure is high [30]. Markus et al. of Karlsruhe Institute of Technology [95] also proposed a special hollow shaft structure made of amorphous alloy, as shown in Figure 13d. This special hollow shaft structure has obvious advantages of low

core loss and good weak magnetic capability, but the eddy current loss of the hollow shaft is high and the slotting process is complex.

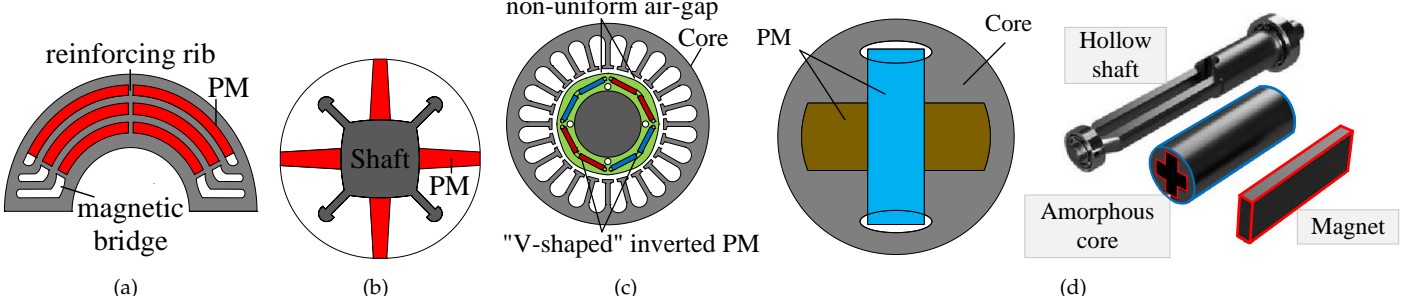

**Figure 13.** Interior permanent magnet structures of high-speed PM motor: (**a**) Interior permanent magnet segmented structure; (**b**) Interior permanent magnet tangential structure; (**c**) Non-uniform air-gap interior permanent magnet structure; (**d**) Hollow shaft permanent magnet special structure.

The rotor wind friction loss of the high-speed PM motor mainly includes air gap loss $p_{air}$, rotor end and thrust collar loss $p_{end}$, axial cooling air loss $p_{axial}$, which can be expressed as [70,96]:

$$\begin{cases} p_{\text{air}} = C_f \pi \rho_a \omega^3 r^4 L_a \\ p_{\text{end}} = 0.5 C_f \rho_a \omega^3 \left(r_2^5 - r_1^5\right) \\ p_{\text{axial}} = 0.67 \pi \rho \left((r + \delta)^3 - r^3\right) v_m u_m \omega \end{cases} \quad (3)$$

where $C_f$ is the friction coefficient, which is related to the rotor surface roughness; $\rho_a$ is the air density; $\omega$ is the rotor angular velocity; $r$ is the rotor radius; $L_a$ is the rotor axial length; $r_1$ and $r_2$ are the inner and outer radii of the rotor end and thrust ring; $v_m$ is the axial cooling air velocity and $u_m$ is the tangential circumferential velocity of the cooling gas at the air gap exit; $\delta$ is the length of the air gap.

Referring to (3), it can be seen that the wind friction loss is mainly affected by rotor speed, thrust disc size, air gap thickness, wind speed and contact surface roughness. Among them, Figure 14 shows the common values of the air gap length for different linear speeds and different rotor outer diameters ($v_c$). From Figure 14, we can see that the rotor outer diameter of high-speed PM motor is mostly taken between 40~170 mm, and the rotor outer diameter of individual ultra-high-speed motor can be taken below 40 mm. The choice of air gap length is mostly concentrated in the range of 1~4 mm.

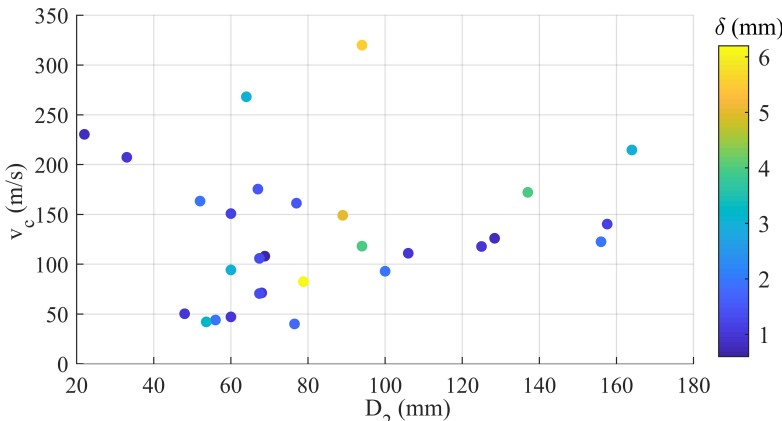

**Figure 14.** Common values of air gap length for different rotor outside diameters and different linear speeds.

## 3. Thermal Management

The goal of thermal management is to accurately solve the temperature rise distribution of the motor and design an efficient heat dissipation system to ensure the efficient and continuous operation of the motor.

Currently, the commonly used calculation methods of motor temperature rise include LPTN, FEM and CFD. LPTN is to solve the temperature distribution by constructing the thermal network of the motor. The calculation speed of this method is fast, but the accuracy is low [13,97]. The FEM can directly profile the motor entity in 2D or 3D modeling and load each loss density and heat transfer condition obtained by empirical methods to obtain a more detailed and intuitive temperature rise distribution than LPTN [98]. The CFD obtains an accurate distribution of temperature rise by jointly modeling the heat transfer entity and the cooling fluid of the motor, which has greater advantages for analyzing and optimizing the heat dissipation structure of the motor, but at the same time, CFD also requires very high computer performance [99]. The comprehensive use of the above schemes to solve the temperature is also widely used in practice.

Efficient heat dissipation is the core of thermal management. The temperature rise of high-speed PM motors mainly comes from the core and windings of the stator and inside the rotor, as shown in Figure 15. In this regard, the cooling system of high-speed motors mostly uses a combination of air-cooled and liquid-cooled structures. As shown in Figure 16a, the air-cooled system is mainly used to achieve heat dissipation between the stator and rotor through air flow, while the liquid-cooled system is mostly used to absorb heat within the stator core section or to exchange heat with the ventilation system to achieve effective heat dissipation [100]. Its inner air channel can also use a closed oil cooling system to replace air to achieve a better cooling effect, as shown in Figure 16b. However, isolation devices are required between the cooling channels and the rotor when oil is passed, which makes the system more complicated, and it is prone to contamination problems caused by oil leakage over a long period of operation [101].

In the ventilation cooling part, the traditional double-end inlet air cooling can achieve better end winding heat dissipation, but the heat dissipation along the stator and rotor axial air gap is poor. Considering that heat dissipation between the stator and rotor is mainly through heat exchange of convective air, a ventilation structure with only one air inlet and one air outlet is applied to force the cooling air to flow along a narrow air gap, and thus the heat dissipation capacity of the air gap is improved [16,60]. Fengge et al. [102] of Shenyang University of Technology designed a hybrid radial and axial cooling structure shown in Figure 17a, which achieves better effect than axial air cooling with double ventilation capacity at the same air speed. Baojun et al. [56] of Harbin Institute of Technology adopted a "hammer slot type" structure based on the semi-closed slot structure, as shown in Figure 17b, which further improves the axial heat dissipation effect of the motor by increasing the axial heat dissipation area with a larger inner slot design. In addition, measures such as arranging cooling fins, creating shallow slots in the rotor sheath and adding appropriate amount of wind spurs on the rotor end face to improve the air flow in the end cavity have also been used to improve the rotor cooling capacity [103,104].

The housing water cooling system has no wind-motion losses, and its structure generally has "Z" axial cooling and circumferential spiral cooling water channel structure. The spiral water-cooling structure has a relatively uniform overall water velocity distribution and is more suitable as a water-cooling structure for high-speed PM motors [105]. Reference [100] pointed out that the cooling efficiency can be improved to some extent by increasing the number of water channels and cooling water flow rate, but this measure also increases the pressure loss in the water channels. In order to circumvent the pressure loss in water pipes and to solve the problem that the water-cooling system is prone to scale, the spray cooling and evaporative cooling methods based on phase change cooling technology have been further applied in recent years [100,106,107]. In addition, the motor cooling solution with additional heat path enhancement also provides ideas for efficient heat dissipation in high-speed PM motors, as shown in Figure 18a,b.

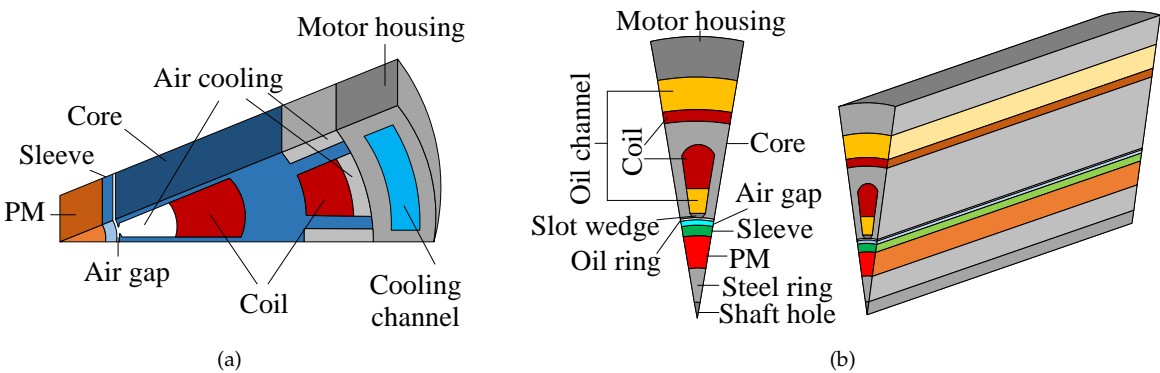

**Figure 15.** Example of temperature distribution of 100 kW 100 kr/min high-speed PM motor.

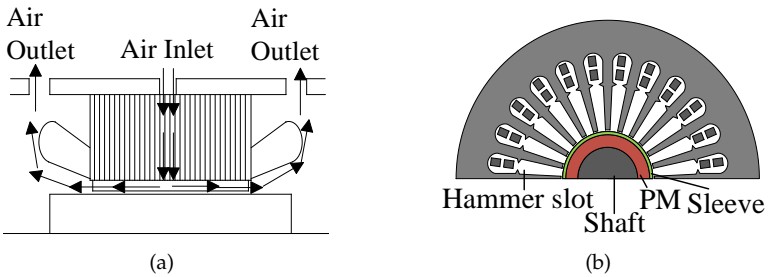

**Figure 16.** Structure of stator opening internal and external cooling channels: (**a**) Hybrid cooling model for toroidal windings; (**b**) Oil cooling model.

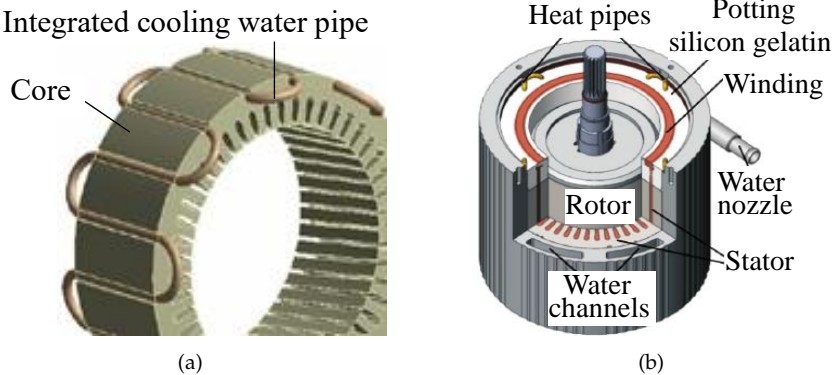

**Figure 17.** Cooling structure: (**a**) The structure of mixed ventilation [102]; (**b**) The structure of "hammer slot type" [56].

**Figure 18.** Motor cooling solution with additional heat path enhancement: (**a**) Stator integrated water pipe [108]; (**b**) Heat pipe and thermal adhesive to enhance heat dissipation [109].

## 4. Rotor Strength and Dynamics

The rotors of the high-speed PM motor are usually designed as slender structures to reduce the centrifugal force on the rotor surface, but the slender structures pose additional challenges to the rotor stiffness and critical speed.

### 4.1. Rotor Strength Analysis

The rotor circumferential speed ($v_c$) of the high-speed PM motor can reach 200 m/s and above, at that time, the high-speed centrifugal force and thermal stress can seriously affect the rotor safety performance. When the rotor is unable to withstand the stresses, it will fail in strength, which will seriously cause the "chamber sweeping" of the stator and rotor and damage the motor [110], as shown in Figure 19.

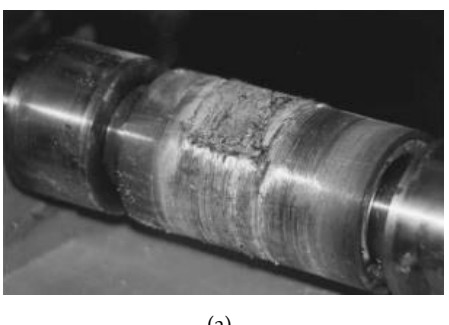
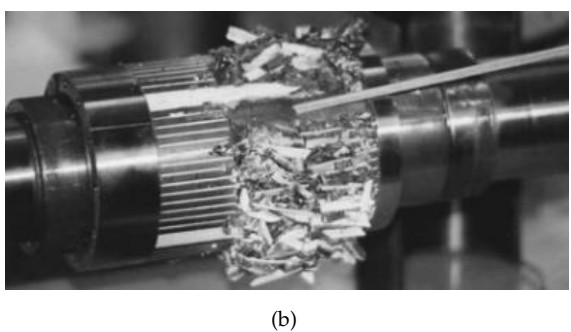

(a)            (b)

**Figure 19.** Destruction of brittle magnets underneath the broken bandage [110].

To protect the rotor, the SPM structure needs to provide a certain amount of preload through the protective material to counteract the tensile stresses generated by the high speed centrifugal force. The pre-stress of the sleeve should be sufficient to counteract the centrifugal force generated by high-speed operation, but should not exceed the compressive degree of PM. The value of pre-stress depends on the temperature, speed, sleeve thickness and the value of interference, where the value of interference is usually less than 0.2 mm [111,112]. Usually, the pre-stress is formed by the interference fit of the alloy sleeve to the PM or by the winding of the fiber composite, as shown in Figure 20a,b. Table 2 shows several rotor bandage materials for high-speed PM motors [94,113–115]. As seen in Table 2, the composite materials have very low density and high strength and resistivity. Alloy materials have significant advantages in terms of heat dissipation capability, temperature stability and stiffness.

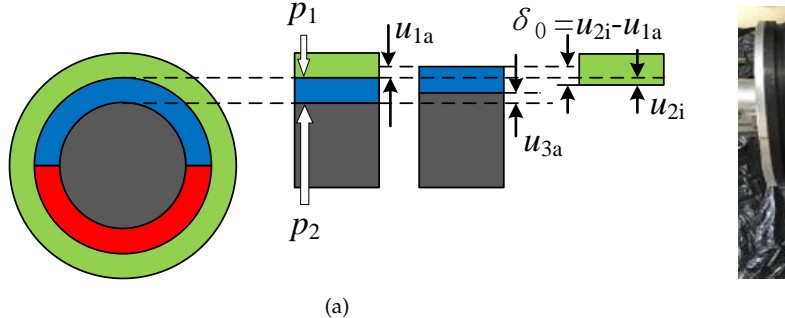
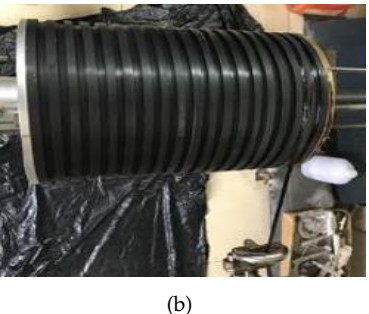

(a)            (b)

**Figure 20.** Two common permanent magnet protection measures: (**a**) Three-layer interference fit. The meanings of the characters in the figure are detailed in [116]; (**b**) Winding process of carbon fiber composites [114].

**Table 2.** Mechanical properties of several rotor bandage materials.

| Parameters | KEVLAR | Carbon Fiber | Fiberglass | Inconel718 |
|---|---|---|---|---|
| $\rho/(\mathrm{g/cm^3})$ | 1.14 | 1.76 | 2.54 | 8.2 |
| $\sigma_{t,\max}/(\mathrm{MPa})$ | 2920 | 3750 | 3447 | 1030 |
| $k/(\mathrm{W}/(m \times K))$ | 0.04 | 5.0 | 1.0 | 11.4 |
| $\rho_t/(\Omega \cdot \mathrm{m})$ | Insulation | $1.5 \times 10^{-5}$ | $4 \times 10^{12}$ | $1.25 \times 10^{-7}$ |

During high-speed operation of the motor, the rotor stresses, strains and displacements will change, and the accurate calculation of each change is critical to the durability of the PM and sleeve. According to the theory related to elasticity mechanics, the equilibrium differential equation of the internal micro-element body of the high-speed rotor subjected to centrifugal force is [117]:

$$\frac{d\sigma_r}{dr} + \frac{\sigma_r - \sigma_\theta}{r} + \rho\omega^2 r = 0 \qquad (4)$$

where $\sigma_\theta$ and $\sigma_r$ are the rotor radial and tangential forces, $r$ is the rotor radius, $\omega$ is the rotor angular velocity, and $\rho$ is the material density.

Refer to (4), the rotor strength analysis of cylindrical PM and ring PM structures with alloy sleeves for different fit models is derived in detail in [116,117]. In engineering practice, metal sleeves are considered to be all isotropic materials, and carbon fibers are considered to be anisotropic materials, hence the strength analysis of the two is significantly different [118]. The detailed derivation of the rotor stress analysis for the carbon fiber tied structure is shown in [114]. When magnets are divided into pieces and produce pole gaps, the gaps need to be filled with non-magnetic material of densities and thermal expansion coefficients closer to those of the magnets to avoid stress concentrations in the sleeves at the edges of the magnetic poles. For this type of structure, C. Liangliang et al. of Zhejiang University [119] derived the strength analytical formula that can be applied under the protection of non-permeable metal sleeve and under the protection of carbon fiber sleeve, respectively. In addition, in order to seek a more accurate theoretical analysis, a generalized plane strain theory based on plane strain was proposed in [120] and verified by FEM, and the results showed that the theory is more suitable for rotors with long shafts.

The equivalent force of the rotor under normal temperature is mainly caused by centrifugal force, and when temperature rise is considered, the rotor will be subjected to uneven thermal stress. The stress distribution of the sleeve for both cases is shown in Figure 21. Among them, Figure 21b take the characteristic of each material having a temperature gradient at high speed into account, which makes the stress distribution closer to the actual situation. To improve the rotor strength, literature [121] indicates that an appropriate increase in the static interference between the PM and the sleeve or increase in the sleeve thickness can improve the preload pressure on the PM and facilitate rotor protection. In addition, measures combining different sleeve materials or mixing magnetic powder with carbon fiber have also been used to improve the bending resistance of the sleeve [92,93].

The IPM structure can simplify the sleeve assembly link, and the magnets are not easily damaged by the direct protection of the rotor core, but the isolated magnetic bridge of the rotor in the IPM structure is subjected to the greatest stress under high-speed operation and is easily damaged, as shown in Figure 22. A moderate increase in the number and thickness of the isolated magnetic bridge is beneficial to improving the mechanical strength of the rotor. For the strength analysis of the IPM structure, due to the complexity of their structure, it is generally necessary to equate the magnets and the protective bridges into equal mass rings and then combine them with finite element modeling for analysis.

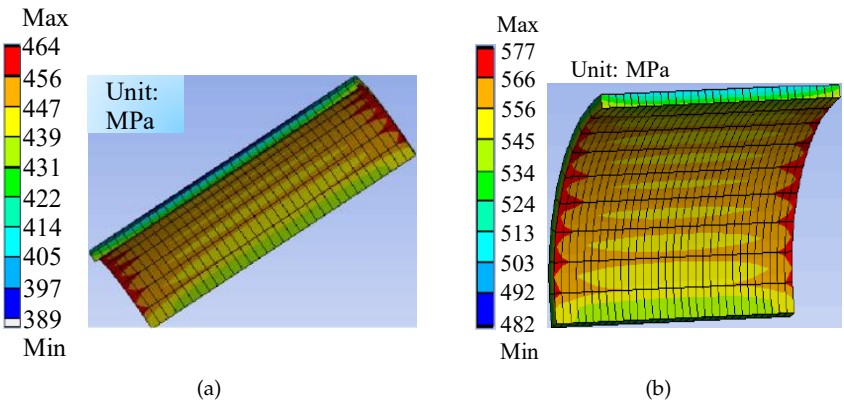

**Figure 21.** The equivalent stress distribution of sleeve at different temperature conditions [122]: (**a**) 20 °C; (**b**) 150 °C.

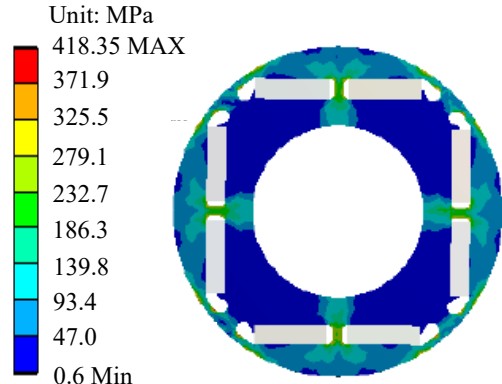

**Figure 22.** Equivalent force distribution of IPM [122].

*4.2. Rotor Dynamics Analysis*

In addition to fulfilling the requirements of rotor strength, stable dynamic characteristics are also pursued for high-speed PM motors, so it is particularly important to perform dynamics analysis. The study of dynamics analysis mainly includes stability analysis, critical speed calculation and unbalanced response.

Stability analysis is closely related to the rotor bearing system, and the main bearings used in high-speed PM motors are ball bearings, oil film bearings, air bearings, and magnetic levitation bearings [82]. Figure 23 shows the distribution of the various bearing schemes reported in [6,7,13,40,50,53,82,93,123–127] relative to the power and speed of the high-speed PM motor.

Table 3 summarizes the application characteristics of several types of bearings. Among them, ball bearings dominate the early stages of high-speed motor bearing supply due to their low cost, high stiffness and ease of assembly. Ball bearings are characterized by the DN number (the product of bearing inner diameter D and rotational speed N) as an indication of high-speed operating capability. By optimizing the topology and improving the lubricating material, etc., Nippon Seiko Co. has developed a high-speed ball bearing with a DN number of more than 1.8 million in 2021. Oil-filled bearings, which support the rotor by the pressure of the oil film formed by the lubricating oil, can achieve lower loss operation than ball bearings, but the additional equipment is complex and the oil leakage problem is prominent [102]. For the demand of high-speed, low loss and high operating life of motors, air bearings and magnetic levitation bearings have gradually become popular research objects. The support stiffness of these non-contact bearings is generally smaller than that of contact bearings, and because of their larger clearances, non-contact bearings have become the main object of stability analysis. The specific study of rotor stability analysis has been discussed in detail in [40,127,128].

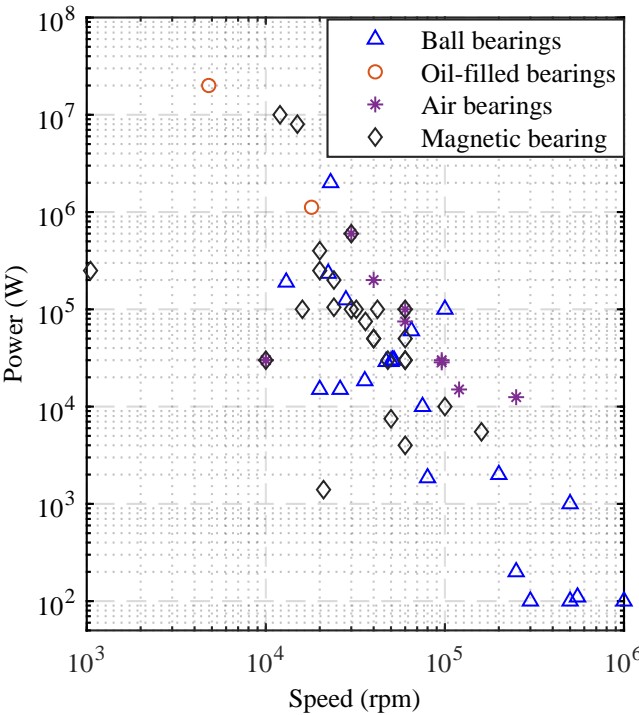

**Figure 23.** Bearing distribution of high-speed PM motor.

**Table 3.** Application characteristics of several types of bearings.

| Bearing Type | Bearing Stiffness (N/m) | Advantages | Disadvantages |
|---|---|---|---|
| Ball bearing | $10^6 \sim 10^9$ | High robustness, small size, low cost and high stiffness. | High bearing loss and short application life at high speed. |
| Oil-filled bearing | $10^7 \sim 10^9$ | Friction coefficient is lower than ball bearings, and it has high impact resistance. | The cooling system is complex and has oil leakage problems. |
| Air bearing | $10^5 \sim 10^8$ | High damping, low friction loss, long service life, compact system, good adaptability. | The load capacity is limited, the dynamic stability is poor, and the performance and processing accuracy of the bearing material are extremely demanding. |
| Magnetic bearing | $10^5 \sim 10^7$ | Great load bearing capacity, no friction loss, good stability. | Complex control system and high cost. |

Due to the presence of an imbalance in the rotor system, when the rotor is running at critical speed, it will excite the natural frequency of the order and cause resonance. To avoid this phenomenon, the motor speed should be kept away from the critical speed. For rigid rotors, the gyroscopic effect on the critical rotor speed can be ignored, and usually, its operating speed is required to run below 80% of the first-order bending critical speed to ensure the safe operation of the rotor. For flexible rotors, the operating speed should be controlled between 1.3 times the first-order critical speed and 0.7 times the second-order critical speed [129]. Typically, non-contact bearing rotors are mostly flexible and more likely to produce dynamic imbalance at high-speed, which in turn generates vibration and affects the stability of the rotor [130].

To facilitate the theoretical resolution of the critical speed, the differential equation for the rotor dynamics can be expressed as [126]:

$$M\ddot{u} + (\Omega G + C)\dot{u} + Ku = F \tag{5}$$

where *M* is the mass matrix of the rotor system; *K* is the stiffness matrix; *C* is the damping matrix; *G* is the gyroscopic matrix; *F* is the excitation force acting on the rotor system; *u* is

the nodal displacement response vector; $\dot{u}$ is the nodal velocity response vector; $\ddot{u}$ is the nodal acceleration response vector; and $\Omega$ is the resonance speed.

Equation (5) states that support stiffness, damping, acceleration and support location all affect the vibration of the system. Among them, the increase in the support stiffness can raise the critical rotor speed, but this method is only suitable for adjustment in a specific range, and beyond this range it will no longer work [120]. In addition, the rotor critical speed also has an inverse relationship with the shaft length and rotor diameter, which also provides a design reference to increase the rotor critical speed and prevent the rotor from developing into a flexible rotor.

The unbalanced response of high-speed PM motors is mainly caused by the unbalanced magnetic pull effect due to the asymmetry of the rotor magnetic circuit. Domestic universities such as Southeast University [131], and Harbin Institute of Technology [132,133] have analyzed the vibration caused by unbalanced rotor response in terms of rotor eccentricity, different magnet thickness and magnetization effects, slot-pole combinations, and air pressure fluctuations to provide a theoretical basis for balancing techniques.

## 5. Multi-Physics Domain Coupling Design

There is a complex coupling relationship between the physical fields of the high-speed PM motor. The interactions between the parameters related to different physical domains are shown in Figure 24. Therefore, the design method of studying a certain part alone can no longer meet the requirements for efficient design of high-speed motors, and the coupled design considering multiple physical domains is more suitable for the design of high-speed PM motors [98].

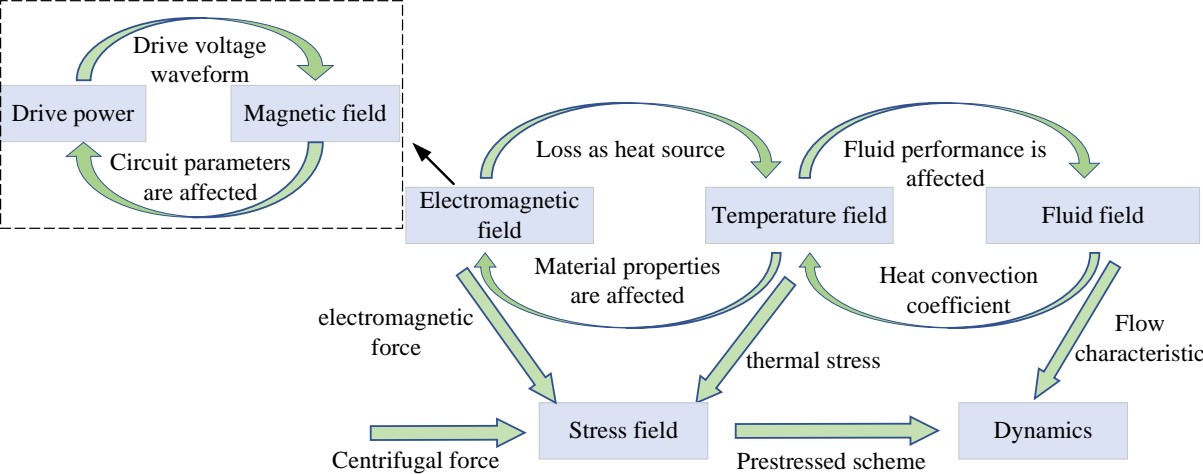

**Figure 24.** Mutual coupling relationship between multiple physical domains.

Coupled design combined with FEM is a common approach in the design of high-speed PM motors today. The study of field-circuit coupling calculation in [134] shows that the field-circuit coupling in the design of high-speed PM motors can ensure that the maximum error of the electromagnetic loss calculations is within 2%. The loss will directly affect the temperature rise. Magnetic-thermal coupling is necessary for the accurate distribution of temperature rise, which mainly including unidirectional coupling and bidirectional coupling. Compared with unidirectional coupling, bidirectional coupling can further take into account the change of relevant electromagnetic parameters with temperature rise and obtain more accurate results, but the calculation of bidirectional coupling is time-consuming [13,25,89,103]. In addition, the coupling between the mechanical stress and the temperature field of the motor under high speed conditions is also of concern, which is discussed in detail and in depth in [122,135].

## 6. Development Trend of High-Speed PM Motor Design Technology

With the increase in demand for high-speed PM motor applications, the efficient design of high-speed PM motors has put forward higher and higher requirements. Based on the analysis of the key technologies of high-speed PM motor design, the research direction of high-speed PM motor design technology is summarized. The main aspects are as follows.

1.  Achieving low loss, high efficiency, high strength, long life, low cost and stable operation of motors is the goal of high-speed PM motor design. Motor losses and rotor strength are related to material properties and structure selection. Therefore, the development of new core materials and innovative structures will still be worth focusing on in the future;

2.  PM material is the most important material for high speed permanent magnet motor. The tensile strength and temperature resistance level of permanent magnet materials are the two major factors that limit the speed and power increase in high-speed permanent magnet motors. Improving these two properties will be a long-term work in the development of permanent magnet materials.

3.  Currently, high-speed PM motors are mostly cooled by a mixture of air-cooled and water-cooled cooling methods, which have a more complex structure and limited cooling effect. With the goal of improving the reliable and efficient operation of motors, innovative motor cooling solutions are also needed in the design phase;

4.  For rotor supports. On the one hand, with the progress of material science and lubrication technology, the engineering problems such as the life of ball bearings in various high-speed occasions need to be further tested and summarized; on the other hand, in domestic, the application and performance evaluation of air bearings and magnetic levitation bearings in practice is still relatively few.

5.  The multi-domain coupling design of high-speed PM motors is extremely important. In terms of design methods, scholars now commonly use FEM or CFD for multi-physics domain coupling design, which can obtain relatively accurate loss, temperature rise and stress distribution, but the calculation process is exceptionally time-consuming. The method of combining FEM and analysis method for integrated and rapid coupling solution of electromagnetic, electrical, mechanical and thermal will be a better choice for the design and analysis of high-speed permanent magnet motors in the future.

## 7. Conclusions

With the application and promotion of permanent magnet motors in the high-speed field, the demand for the efficient design of high-speed PM motors has been pushed to a new height. The design process of high-speed PM motors needs to face problems of high losses, high temperature rise and more complex mechanical characteristics, which must be achieved by closely considering the interaction of the electromagnetic, temperature and mechanical fields. This article briefly describes the design strategies of high-speed PM motors, and the research and application of new materials and structures in high-speed PM motors. The selection basis of the major parameters of high-speed PM motors is outlined, and the characteristics and applications of different structures and materials are summarized. The calculation method of temperature rise and the efficient management measures of temperature at the present stage are outlined. Common protection methods for permanent magnets and strength analysis methods for different rotor structures are discussed. The development and applications of different bearings are overviewed, and solutions for reducing the resonance and unbalance response of the rotor are discussed. The mechanism of the interaction between the relevant parameters in each physical domain is summarized. Finally, the future direction of the efficient design of high-speed PM motors is foreseen. Permanent magnet motors have great potential for development in the high-speed field, so the material development, structural innovation and design strategies of high-speed PM motors require further in-depth research.

**Author Contributions:** Conceptualization, Q.S., Z.Z., S.L., X.L., T.W., X.H. and J.Z.; methodology, Q.S., Z.Z. and S.L.; software, Q.S. and Z.Z.; validation, Q.S., Z.Z., S.L., X.L., T.W., X.H. and J.Z.; formal analysis, Q.S., Z.Z. and S.L.; writing—original draft preparation, Q.S., Z.Z. and S.L.; writing—review and editing, Q.S., Z.Z. and S.L.; supervision, Q.S.; project administration, Q.S.; funding acquisition, Q.S. All authors have read and agreed to the published version of the manuscript.

**Funding:** This research was funded by National Natural Science Foundation of China, grant number 51607017 and Scientific Research Foundation of Chongqing University of Technology, grant number 2019ZD78.

**Institutional Review Board Statement:** Not applicable.

**Informed Consent Statement:** Not applicable.

**Data Availability Statement:** Not applicable.

**Conflicts of Interest:** The authors declare no conflict of interest.

## Abbreviations

The following abbreviations are used in this manuscript:

| | |
|---|---|
| DC | Direct Current |
| AC | Alternating Current |
| SPM | Surface-mounted Permanent Magnet |
| IPM | Interior Permanent Magnet |
| IM | Induction Motor |
| PM | Permanent Magnet |
| PMSM | Permanent Magnet Synchronous Motor |
| SRM | Switched Reluctance Motor |
| SMC | Soft Magnetic Composite |
| Back-EMF | Back Electromotive Force |
| LPTN | Lumped-Parameter Thermal-Network |
| FEM | finite element method |
| CFD | Computational Fluid Dynamics |
| Litz | Litzendraht |

## Appendix A

**Table A1.** Data resources for different dimensional parameters of high-speed permanent magnet motors with different structures.

| Rated Power (kW) | Rated Speed (krpm) | Efficiency | Poles | Stator Slots | Rotor Structure | $D_{i1}/D_1$ (mm) | $L_{ef}$ (mm) | Iron Loss (W) | Copper Loss (W) | Air Gap Height (mm) | Reference |
|---|---|---|---|---|---|---|---|---|---|---|---|
| 300 | 15 | - | 2 | 36 | SPM | 160/350 | 280 | 2225 | 1137 | 2 | [8] |
| 132 | 18.75 | 0.98 | 4 | 24 | IPM | 130/220 | 130 | - | - | 0.8 | [11] |
| 120 | 24 | 0.95 | 2 | 24 | IPM | 102/200 | 112 | - | - | 4 | [12] |
| 75 | 60 | 0.98 | 2 | 24 | SPM | 56/126 | 128 | 230 | 294 | 2 | [13] |
| 200 | 20 | - | 2 | 24 | SPM | 135/295 | - | 2500 | 1200 | - | [16] |
| 100 | 100 | - | 2 | 24 | SPM | 79/150 | 165 | - | - | 1 | [19] |
| 15 | 30 | - | 2 | 18 | SPM | 60/130 | 70 | - | - | - | [20] |
| 100 | 50 | - | 2 | 36 | SPM | 70/200 | 160 | 814 | 458.2 | 1.5 | [22] |
| 150 | 30 | - | 2 | 24 | SPM | 125/250 | - | 1291.8 | - | - | [14] |
| 150 | 20 | - | 4 | 36 | SPM | 158/336 | 120 | 1684 | 165.5 | - | [25] |
| 250 | 67 | - | 2 | 24 | SPM | -/368 | 155 | 11990 | - | 5.5 | [26] |
| 220 | 18 | 0.97 | 2 | 24 | SPM | 127/205 | 308 | 434.4 | 121.6 | 1 | [27] |
| 200 | 20 | - | 2 | 24 | SPM | 135/295 | 200 | 2500 | 1200 | - | [60] |
| 120 | 17.75 | - | 4 | 24 | SPM | 104/195 | 130 | 963 | 280 | 2 | [15] |
| 100 | 40 | - | 4 | 36 | SPM | 80/180 | 138 | - | - | - | [18] |
| 10.5 | 100 | - | 2 | 24 | cylindrical | 38/104 | 33 | - | - | - | [21] |
| 35 | 135 | - | 2 | 12 | cylindrical | 43.4/138 | 62 | - | - | - | [24] |
| 100 | 32 | - | 4 | 24 | SPM | 99/230 | - | - | - | 5 | [28] |
| 45 | 48 | - | 4 | 24 | IPM | 62.4/120 | 100 | 642.7 | 200.5 | 1.2 | [30] |
| 220 | 18 | - | 4 | 24 | SPM | 75/120 | - | - | - | - | [33] |
| 20 | 30 | - | 2 | 36 | SPM | 66/130 | 90 | - | - | 3 | [34] |
| 10 | 100 | - | 2 | 24 | SPM | 40/104 | 36 | - | - | - | [35] |
| 15 | 120 | - | 2 | 12 | cylindrical | 35/120 | 50 | 124 | 520 | 1 | [36] |

**Table A1.** *Cont.*

| Rated Power (kW) | Rated Speed (krpm) | Efficiency | Poles | Stator Slots | Rotor Structure | $D_{i1}/D_1$ (mm) | $L_{ef}$ (mm) | Iron Loss (W) | Copper Loss (W) | Air Gap Height (mm) | Reference |
|---|---|---|---|---|---|---|---|---|---|---|---|
| 800 | 25 | - | 4 | 24 | SPM | 170/360 | - | - | - | 3 | [37] |
| 6.2 | 15 | 0.96 | 2 | 24 | SPM | 62/120 | 110 | 70 | 47.6 | 1 | [39] |
| 80 | 80 | - | 2 | 24 | SPM | 70/182 | 100 | 617 | 325 | 3 | [41] |
| 100 | 60 | - | 2 | 12 | SPM | 100/124 | 102 | - | - | - | [42] |
| 100 | 20 | - | 2 | 18 | SPM | 108/250 | 104 | 700.3 | 668 | 1 | [43] |
| 150 | 17 | - | 4 | 36 | SPM | 160/350 | 140 | 1135.3 | 432.6 | 1.2 | [90] |
| 7.5 | 50 | - | 2 | 18 | SPM | 55/116 | 50 | - | - | - | [46] |
| 20 | 100 | - | 2 | 12 | cylindrical | 34.8/- | - | - | - | - | [48] |
| 75 | 24 | 0.98 | 2 | 24 | SPM | 145/240 | 100 | 660 | 545 | 4 | [49] |
| 40 | 40 | - | 2 | 24 | SPM | -/- | - | - | - | - | [52] |
| 7.5 | 15 | 0.94 | 2 | 18 | SPM | 60/130 | 70 | 156.6 | 64.2 | 3.2 | [97] |
| 15 | 30 | - | 4 | 18 | SPM | 70/130 | 110 | 274 | 102 | 1.3 | [89] |
| 10 | 24 | - | 4 | 24 | SPM | 60/115 | 70 | - | - | - | [77] |
| 10 | 10 | - | 4 | 12 | SPM | 80/160 | 78 | 133.4 | 87.3 | 1.75 | [58] |
| 1120 | 18 | - | 4 | 27 | SPM | -/550 | 400 | - | - | 3 | [102] |
| 5 | 20 | 0.948 | 4 | 18 | IPM | 70/130 | 110 | 215.7 | 158.8 | 1.3 | [136] |
| 7.5 | 15 | 0.95 | 4 | 18 | SPM | 60/- | 70 | - | - | 2 | [137] |
| 15 | 20 | 0.96 | 4 | 18 | IPM | 70/- | 110 | - | - | 1 | [137] |
| 5 | 20 | 0.94 | 4 | 24 | SPM | 50/100 | 55 | 32 | - | 1 | [138] |
| 100 | 50 | - | 2 | 36 | SPM | -/200 | 160 | - | - | 1.5 | [111] |
| 30 | 20 | - | 6 | 36 | IPM | 91.2/154.2 | 135.9 | - | - | 6.2 | [139] |
| 2 | 200 | - | 2 | Slotless | cylindrical | 23.6/35 | 13 | 9.85 | 20.3 | 0.8 | [140] |
| 40 | 40 | - | 4 | 36 | SPM | 80/135 | 125 | - | - | 1.5 | [141] |
| 5 | 30 | - | 4 | 24 | IPM | 70/150 | 59 | - | - | 0.6 | [142] |
| 100 | 32 | - | 4 | 24 | SPM | -/245 | - | - | - | - | [143] |
| 250 | 65 | - | 2 | 36 | SPM | 105/240 | 140 | 5009 | 1025.7 | 5.5 | [144] |

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
