# Peer review of "Design and Analysis of the High-Speed Permanent Magnet Motors: A Review on the State of the Art"

_machines, doi:10.3390/machines10070549_

Round 1

Reviewer 1 Report

Summary

This article reviews the state of the art of permanent magnet synchronous motors, focusing on some of the critical aspects of permanent magnet motor design: thermal management, rotor strength, dynamics analysis and multi-physics domain coupling.

General comments

This article provides an interesting state-of-the-art review of high-speed permanent magnet machines.

It shows an abundant bibliography as well as an extensive list of designed motors with their corresponding bibliographical references. Together with this list, the most interesting parameters are shown in order to know the characteristics of each machine.

The article focuses on the main challenges in the design of high-speed motors, mainly: losses vs. temperatures and mechanical design of the rotor.

With regard to the section on losses, more detail is lacking on the distribution of losses in the motors (copper, iron and mechanical). It would also be interesting to show the influence of iron losses, due to the high frequencies of the currents and their corresponding harmonics. It would also be interesting to show the losses in the conductors due to the proximity effect.

Another section that is not covered is the finite element analysis of the problems shown, which is a crucial issue for good motor design.

Specific comments

-          Line 118: It would be interesting to indicate the thicknesses of magnetic sheet being used.

-          Figure 6: Which is the thickness of 35W270 magnetic steel?

-          Figure 12: Try to improve image quality.

-          Section 2. Electromagnetic design:

o   It is possible to show the losses distribution (copper and iron losses) in some of the machines of Annex A?

o   A key aspect of electromagnetic design is the calculation of iron losses, due to the high frequencies at which we work. Also, the losses in copper due to the proximity effect. It would be interesting to show data on this.

-          Section 2.2: It would be interesting to comment on the air gap height of the motors as a function of the outer diameter and the peripheral speed of the rotor.

-          Section 3. Thermal Management: In this section, it would be very interesting to have a graph showing the current densities obtained for: the motors shown in Annex A and the current densities obtained with the different cooling systems shown.

-          Section 4. Rotor Strength: It would be interesting:

o   Introduce references of the materials used for the manufacture of the bandage.

o   the value of the sleeve pre-stressing is not explained in detail

-          Figure 17, (b) The equivalent stress distribution derived from the thermal-structural coupling of the sleeve in thermal operation: What temperature is the sleeve working?

-          Figure 18: Try to improve image quality.

Reviewer 2 Report

The authors provide a valuable review in the field of design and analysis of the high-speed permanent magnet motors, however I have some recommendations:

- please re-check the English, in some sentences the subject is missing, in some the verb, in some wrong person is used;

- in the introduction section the authors should stress out more the differences among the named motor types, especially their operating principles;

- line 78, the "A" letter seems to be placed lower compared to other letters, please correct;

- Figure 5 - the a) and b) cases do not have coils sketched, please add;

- Figure 7 - the figure is not very clear considering broader readers base, the current direction representation would be nice also, please modify and add;

- Equation 4 - in the first expression, should the "r" in the denominator be an index?

I also recommend to extend the section 6 of the paper. Is there some new and groundbreaking knowledge that has to be pointed out in this section? This information would be nice.

Author Response

请看附件

Reviewer 3 Report

First of all, thank you for your article.

I have a few questions about the author’ article.

Q1. First of all, Reviewers are very interested in this paper. This article covers a variety of topics and is informative.

Q2. In line 17, what does r/min √kW mean? This expression is rarely used. Do you mean speed per kW? Also, the sentences in lines 16-17 are strange.

Q3. The reviewer does not know what Figure 1, b) means. Of course, it is good to show the capacity distribution of high-speed motors in the article, but it is recommended to classify them as in Fig. 1, a).

Q4. How about deleting the data of the outer rotor, twin rotor, and twin stator in Fig. 2, a)? In Figure 2, a), the quantity of related data is also small and confuses the meaning of existing data. It is recommended to show mainly SPM, IPM, and Solid PM. 

Q5. Reviewer will understand the meaning of "Solid PM" mentioned in this article. But Reviewer don't know if this is the right way to express it. Referring to other papers, it is recommended that you choose an appropriate word. (Fig 2, a~b))

Q6. On line 41, the term “object” is not appropriate. It is recommended to change to another term to fit the sentence.

Q7. In line 98, there are several main reasons for choosing the smaller number of poles on high-speed rotating machines. Among them, it is to prevent scattering at high speed and, in particular, to reduce eddy current loss of permanent magnets. It is good to highlight the advantages.

Q8. In Figure 6, the distinction between the coating part and the binder part is not clear. It is recommended that the figure be modified.

Q9. Fig. 10 d) is not exactly what the structure is. Revision is necessary whether to change the picture or reinforce the content. 

Q10. Reviewer think "Rotor Dynamics Analysis" is the most important for high-speed rotating machines. Figure 19 provided by the author is also very important. In order to increase the understanding of the article, it would be good to add the characteristics of each type of bearing. The sentences on lines 350-360 refer only to fragmentary features of the bearing.

Q11. Overall, the font of the text inserted in the picture is large. Correction is required. And the sentence is a bit strange. A general inspection is required.
